# EgoQR: Efficient QR Code Reading in Egocentric Settings

## Abstract

QR codes have become ubiquitous in daily life, enabling rapid information exchange. With the increasing adoption of smart wearable devices, there is a need for efficient, and friction-less QR code reading capabilities from Egocentric point-of-views. However, adapting existing phone-based QR code readers to egocentric images poses significant challenges. Code reading from egocentric images bring unique challenges such as wide field-of-view, code distortion and lack of visual feedback as compared to phones where users can adjust the position and framing. Furthermore, wearable devices impose constraints on resources like compute, power and memory. To address these challenges, we present EgoQR, a novel system for reading QR codes from egocentric images, and is well suited for deployment on wearable devices. Our approach consists of two primary components: detection and decoding, designed to operate on high-resolution images on the device with minimal power consumption and added latency. The detection component efficiently locates potential QR codes within the image, while our enhanced decoding component extracts and interprets the encoded information. We incorporate innovative techniques to handle the specific challenges of egocentric imagery, such as varying perspectives, wider field of view, and motion blur. We evaluate our approach on a dataset of egocentric images, demonstrating 34% improvement in reading the code compared to an existing state of the art QR code readers.

## 1 Introduction

Quick Response (QR) codes have revolutionized information exchange, enabling rapid data transfer across various applications. By encoding diverse data types such as URLs, plain text, and contact information, QR codes have achieved widespread adoption, becoming ubiquitous in daily life from marketing to payment systems Shin et al. (2012); Tiwari (2016). QR code decoding on smartphones relies on user interaction, allowing for adjustments in focus, positioning, and orientation until a successful scan is achieved. In fact, in these devices users receive real-time feedback, allowing them to optimize the scanning conditions actively. However, the growing adoption of smart wearable devices, requires building a frictionless QR code reading system with minimal feedback from the user.

Integrating QR code functionality into wearable devices poses significant challenges. Unlike smartphones, wearable devices capture images in a single shot, without user adjustment or feedback. User expectations also differ significantly between smartphones and wearable devices. Smartphone users are accustomed to actively framing the QR code within the camera view, whereas with wearables, users expect detection and decoding of QR codes within their field of vision. This expectation of "seeing what I see" presents unique challenges for egocentric captured images. Users may view QR codes from various angles and distances, resulting in distorted perspectives unlike the typically straight-on view achieved with smartphones. Natural head movements can introduce motion blur, which the algorithm must compensate for. The QR code may not always be in the primary focus area of the image, requiring more robust detection methods. Varying lighting conditions can affect the binarization process in the decoding process, causing uneven illumination. Additionally, wearable devices must handle a wide range of lighting conditions and backgrounds as users move through different environments, introducing environmental variability. This shift in user expectations presents both opportunities and challenges. While it promises a more intuitive and frictionless interaction, it

also demands more robust and adaptable QR code reading algorithms capable of handling suboptimal imaging conditions (Figure 8). Additionally, running the QR code reader on edge devices poses unique challenges due to limited resources, including processing power, memory, and battery life, requiring efficient algorithms to balance performance and resource utilization.

Existing QR code reading systems struggle to address these challenges, where not only they are not optimized to run in limited resource settings but also they not meant to handle egocentric images, thus resulting in reduced code reading success; leaving efficient, real-time solutions remain elusive. In light of these challenges and evolving user expectations, this paper presents a novel, lightweight algorithm designed to address these specific challenges, aiming to enhance the QR code reading experience in egocentric settings. Our approach focuses on efficient yet more robust detection and decoding components that can operate on high-resolution images with minimal power consumption and added latency. Specifically, we developed a light weight code detection model paired with a enhanced decoding algorithm on top of ZXing Crossing") library and could improve the decoding success rate by 34% compared to state of the art code readers on a ego-centric dataset collected for the purpose of this work.

## 2 PREVIOUS WORKS

Early QR code detection methodologies relied on traditional computer vision techniques, which primarily sought to identify distinct geometric features of QR codes. One prominent method proposed a fast approach for detecting QR codes in arbitrarily acquired images, aiming to minimize computational demands while maintaining a high detection rate Belussi & Hirata (2011). This work laid the foundation for subsequent research efforts by providing a solution to detect QR codes in real-world, uncontrolled conditions. In a related effort, a component-based approach was introduced to further enhance detection accuracy by breaking the QR code into recognizable parts—such as finder and alignment patterns—thereby improving detection rates in environments with complex backgrounds Belussi & Hirata (2013a). Despite their effectiveness under controlled conditions, these methods had difficulty generalizing to diverse environments due to their reliance on manually engineered features that struggled with variability in image capture conditions.

The introduction of machine learning approaches revolutionized QR code detection by enhancing robustness and reducing dependency on handcrafted features. The use of convolutional neural networks (CNNs) for automating feature extraction marked a significant step forward Chou et al. (2015b). By allowing models to learn features directly from data, these techniques avoided the limitations of traditional methods, improving adaptability to distortions and variations in lighting or perspective. In another advancement, a local feature-based QR code detection approach was developed to emphasize the extraction of unique patterns, helping to enhance detection in cluttered or challenging scenes Belussi & Hirata (2013b). Complementing these developments, a focus on QR code detection in high-resolution images Szentandrási et al. (2012) provided important insights into balancing detection speed with improved accuracy, demonstrating how streamlining the pipeline could yield faster results without sacrificing performance.

Further progress came through optimization strategies aimed at faster and more reliable detection. An algorithm designed for rapid detection reduced the computational burden while maintaining robustness, making it particularly useful for real-time applications Zhang et al. (2017). Enhancements based on the parallel line features of QR codes were also introduced, showing how geometric properties could be leveraged to strengthen detection accuracy across varying conditions Klimek & Vámossy (2013). Another innovative approach used BING and AdaBoost-SVM techniques to boost performance in detection Yuan et al. (2019), while a binary large object-based detection method focused on effectively segmenting image components to enhance accuracy in uncontrolled settings Lopez-Rincon et al. (2017). These strategies collectively contributed to making QR code detection faster and more adaptable to different environments.

Challenges associated with low-resolution images have long been a significant limitation in QR code recognition, as they often result in poor detection rates. To mitigate these issues, super-resolution techniques have been applied to improve the quality of low-resolution QR code images, thus enhancing detection reliability. A deep learning-based super-resolution method was proposed to enhance QR code clarity, enabling detection systems to perform effectively even when presented with low-quality inputs Shindo et al. (2022). Another similar effort utilized the inherent structural features

of QR codes to reconstruct high-quality images from low-resolution captures, thus boosting the effectiveness of subsequent detection steps Kato et al. (2011). These approaches have demonstrated the potential for pre-processing techniques to play an important role in enhancing recognition in challenging situations, thereby broadening the practical applicability of QR code detection systems.

The application of deep learning techniques to QR code detection continued to evolve, with comprehensive evaluations demonstrating the superiority of these models over traditional methods. A study evaluating various deep learning architectures showed that properly trained models could significantly outperform conventional approaches under a wide array of image conditions Blanger & Hirata (2019). Additional advancements in the use of CNNs sought to address issues of distortion and extreme angles, thereby extending the practical use of QR code detection systems to cases involving perspective challenges Chou et al. (2015a). These improvements were pivotal for enabling detection systems to function accurately regardless of camera angles or image orientations.

The integration of sophisticated detection models further enhanced QR code detection systems. Faster-RCNN, combined with Feature Pyramid Networks (FPNs), was used to detect QR codes in complex environments Peng et al. (2020). FPNs allowed the model to extract and process features across multiple scales, thus improving detection performance across a wide variety of QR code sizes and environments with significant background clutter. This multi-scale capability was important for differentiating QR codes from other visual elements. Additionally, precise localization of QR codes using deep neural networks was explored, which helped address the challenge of distinguishing QR codes from other patterns that might exist in natural scenes Grósz et al. (2014).

Several foundational libraries have supported QR code detection and processing. Libraries such as ZXing Crossing"), ZBar ZBar, and Quirc Quirc") have been pivotal in providing robust tools for reading QR codes, each offering unique strengths. ZXing stands out for its versatility and wide adoption, ZBar is notable for its lightweight design and ability to detect multiple codes within a single image, while Quirc focuses on efficiency and high-performance scanning. OpenCV's QRCodeDetector opencv has offered an accessible, integrated approach to QR code detection by including functions for enhancement, localization, and decoding. Additionally, Dynamsoft's SDK Reader provides a commercial-grade solution for recognizing QR codes, focusing on optimized algorithms for increased reliability.

## 3 ARCHITECTURE

Figure 1 depicts the overall architecture of our flow. The process commences with image downsizing for efficient QR code detection. This module identifies QR code bounding boxes within the image and rescales them to their original size. Subsequently, the code reader extracts each QR code and applies necessary image processings to the code patch, then attempting decoding until successful. The decoded codes, potentially multiple, are then forwarded to the fulfillment module. Leveraging user input signals, such as pointing gestures, this module identifies the most relevant QR code. Finally, the selected code is transmitted to the application for display to the user.

### 3.1 DETECTION

To efficiently detect QR code patches, we employ a detection model that processes a $576 \times 432$ thumbnail image, optimized for memory and latency constraints. Our approach leverages the Faster R-CNN framework Ren et al. (2015), with tailored anchor box distributions specifically designed for QR code detection. The model is trained on a dataset of approximately 15,000 images, achieving 94% recall and 95% precision at a 0.5 IOU threshold.

However, two primary challenges arise in QR code detection. Stylistic QR codes pose a significant challenge due to their limited representation in our training data. While our model excels at detecting traditional square-shaped QR codes with black patterns, it exhibits reduced robustness when encountering stylistic variations. Figure 2.a illustrates examples of these cases. Small QR codes also prove difficult to detect due to the egocentric nature of the images. Specifically, QR codes occupying $15 \times 15$ pixels or less in the thumbnail image are challenging to identify. Notably, the majority of false positives arise from small patches. Figure 2.b demonstrates the scale of a 0.8-inch QR code within the captured image, highlighting the detection challenges.

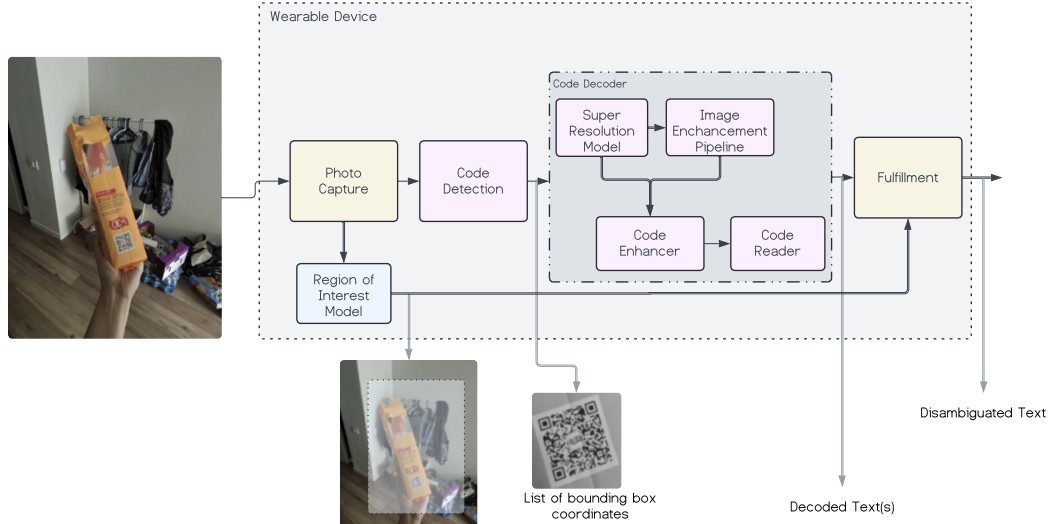

Figure 1: Overall architecture of the code reading flow from the phone taken from user's command to final fulfilment.

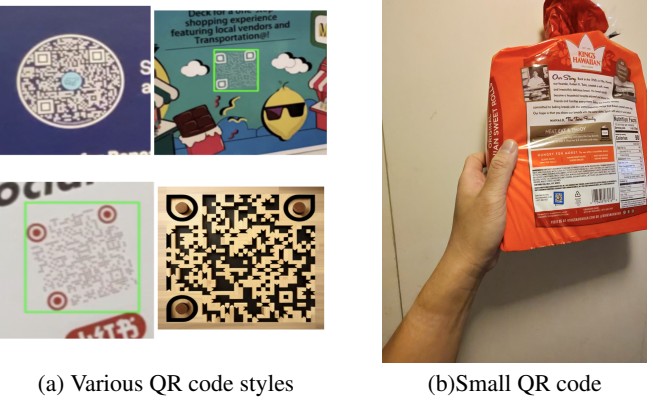

(a) Various QR code styles        (b)Small QR code

Figure 2: QR code detection examples

## 3.2 DECODING

While detection focuses on identifying QR code patterns, decoding relies on precise pixel-level information, making it more susceptible to image quality issues. To improve decoding success rates, we developed a multi-trial preprocessing algorithm that enhances the image quality of detected QR codes. After applying our high-recall detection model to the full image, we extract each detected QR code from the high-resolution grayscale image, adding a margin around the code. We then apply the following image enhancement techniques iteratively:

- Color Inversion: We process both the original and inverted color versions of the code. The inversion is performed as:

$$I_{inv}(x, y) = 255 - I(x, y) \tag{1}$$

  where I(x,y) is the intensity of the pixel at (x,y).

- Multi-scale Processing: We process the code at four different scale values, including upscaling and downscaling.

- Unbalanced Contrast Handling: a) OTSU Binarization: We use Otsu's method Otsu (1979) to find the optimal threshold ($t$) that minimizes the intra-class variance:

$$\sigma_{w(t)}^2 = w_1(t)\sigma_1^2(t) + w_2(t)\sigma_2^2(t) \tag{2}$$

where $w_1$ and $w_2$ are the probabilities of the two classes separated by threshold $t$, and $\sigma_1^2$ and $\sigma_2^2$ are the variances of these two classes. b) CLAHE (Contrast Limited Adaptive Histogram Equalization Pizer et al. (1987)): We apply CLAHE with two clip limit values ($\beta$) to enhance local contrast:

$$P_{clip}(g) = min(P(g), \beta * (1/N) * \sum P(i)) \tag{3}$$

where $P(g)$ is the histogram count for gray level $g$, and $N$ is the total number of gray levels.

– Morphological Operations: We apply dilation and erosion to enhance the QR code structure, where ($K$ is the structuring element):
Dilation: $(I \oplus K)(x,y) = \max\{I(x - x', y - y') + K(x', y') | (x', y') \in K\}$
Erosion: $(I \ominus K)(x,y) = \min\{I(x + x', y + y') - K(x', y') | (x', y') \in K\}$

– Super Resolution: For small QR code patches ($< 192 \times 192$ pixels), we apply a super-resolution model to increase resolution and potentially improve decoding success (details below). The super-resolution process can be represented as $I_{HR} = f_{SR}(I_{LR})$ where $f_{SR}$ is our super-resolution model.

After each enhancement step, decoding is attempted. The process terminates once decoding is successful or all enhancement techniques have been applied. It's important to note that while some of these steps may have overlapping effects, we maintain this iterative and ordered process to minimize overall latency. All steps except the last one are based on fast computer vision algorithms. The super-resolution step, requiring approximately 20ms to run, is applied last due to its higher computational cost.

### 3.2.1 SUPER RESOLUTION

We utilize a custom implementation of the LRSRNGankhuyag et al. (2023) model, adapted for our specific use case, to perform super-resolution. This was chosen for its favorable latency characteristics. The model was trained on approximately 700,000 pairs of low-resolution to high-resolution QR code patches, with the high-resolution patches primarily sourced from MetaClip datasetsXu et al. (2024). To generate low-resolution images, we used simulation techniques to mimic camera noise. The super-resolution step improved our decoding success rate by 2.6% as shown later. The super-resolution model aims to learn a mapping function $f_{SR}$ that minimizes the difference between the generated high-resolution image and the ground truth:

$$\arg \min_{\theta} E\left[L\left(f_{SR}\left(I_{LR}; \theta\right), I_{HR}\right)\right] \tag{4}$$

where $L$ is a loss function (e.g., mean squared error), $I_{LR}$ is the low-resolution input, $I_{HR}$ is the high-resolution ground truth, and $\theta$ are the model parameters. An example QR code passed through this model is shwon in figure 3.

This comprehensive approach to image enhancement, combining traditional computer vision techniques with advanced machine learning models, allows us to significantly improve QR code decoding success rates in challenging egocentric scenarios.

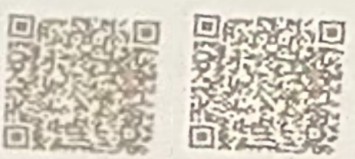

Figure 3: Super resolution input (left: not decodable) and output (right: decodable).

### 3.3 FULFILLMENT

### 3.3.1 HANDLING DISAMBIGUATION

In egocentric settings, due to the abundance of QR codes in the wild, it can be challenging to selectively scan specific QR code(s) that users may be primarily interested in. To address this, we employ

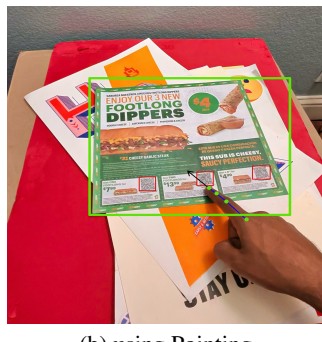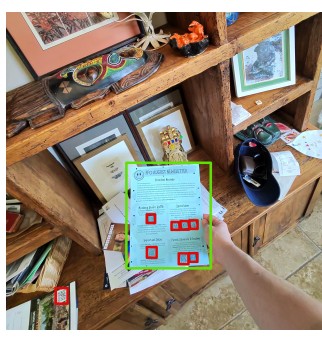

|(a) using ROI|(b) using Pointing|(c) Edge Case with Multiple Codes|

Figure 4: Examples of QR Code Disambiguation.

a disambiguation technique to present the top-1 QR code to the user after successful detection and decoding. We leverage the Region Of Interest (ROI) Detection module from Lumos Shenoy et al. (2024) which provides us with a bounding box around the most salient ROI of the image as well as a "pointing vector" based on the linear extrapolation of index finger joint keypoints, if index-finger pointing is detected. This additional information enables us to:

1. Prioritize QR codes within the detected ROI, assuming codes within the ROI (e.g., a QR code on a handheld flyer) are more relevant than background codes. We shortlist candidates by checking inclusion within the ROI bounding box.

2. Select the QR code directly pointed to by the user, utilizing line-box intersections between candidate patches and the "pointing vector" to find the closest intersecting patch to the vector origin.

To handle edge cases where ROI detection is uncertain or yields 0 or multiple QR codes, we fall back to selecting the largest candidate based on pixel area.

### 3.3.2 HANDLING READING ERRORS

When QR code detection or decoding fails, we provide users with relevant feedback to facilitate retrying from a different angle or closer proximity. If no QR codes are detected in the image, we notify the user that no QR code was found and prompt them to try again. If at least one QR code is detected but decoding fails, we notify the user that the QR code image is unclear and suggest retrying.

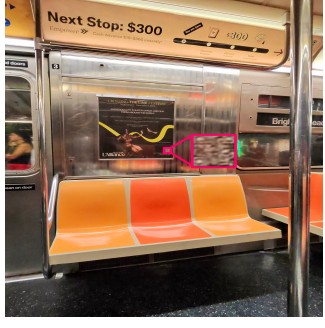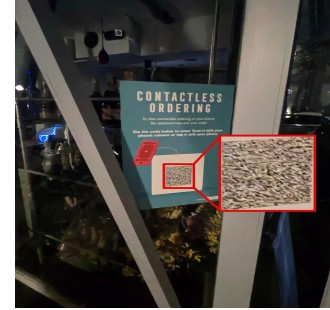

|(a) Very Small QR Code|(b) Unclear QR Code|

Figure 5: Examples of QR Code Reading Errors.

# 4 EXPERIMENTAL RESULTS

## 4.1 EXPERIMENTAL SETUP

### 4.1.1 BENCHMARK DATASET

The majority of existing QR code datasets are not representative of egocentric scenarios, which present unique challenges for evaluating algorithms developed for wearable devices. To address this gap and evaluate our algorithm in realistic conditions, we collected a novel dataset of egocentric images captured "in the wild". Our data collection process was designed to capture a diverse range of real-world scenarios. Participants captured egocentric images throughout their daily activities under various lighting conditions, both indoors and outdoors. To avoid biases towards unrealistic or overly optimized scenarios, we provided no specific instructions regarding QR code placement, lighting conditions, or image composition. This ensured that our dataset accurately represents the challenges faced in actual use cases of egocentric QR code reading. The resulting dataset comprises 528 total images, featuring both single and multiple QR codes per image. In total, these images contain 697 unique QR codes, providing a robust foundation for evaluating our algorithm's performance across diverse scenarios. Figure 6 presents a selection of images from our dataset, illustrating the variety and challenges present in egocentric QR code scanning.

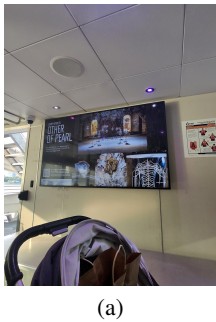 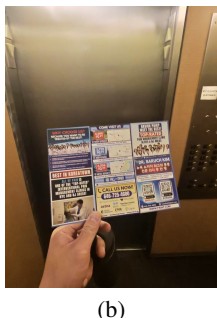 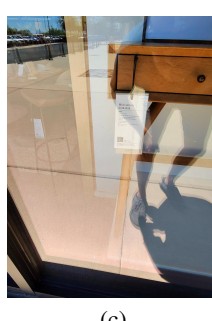 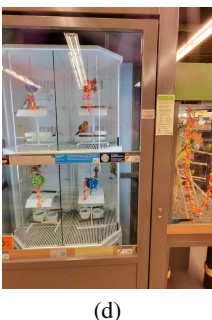

(a)         (b)         (c)         (d)

Figure 6: examples from our collected egocentric QR code benchmarking dataset

As evident from these examples, egocentric settings present unique challenges for QR code detection and decoding. Our dataset captures a range of difficulties commonly encountered in wearable device scenarios, including motion blur due to natural head movement, focus variations where QR codes are not always at the image's focal point, and diverse lighting conditions across indoor and outdoor environments. Additionally, the egocentric perspective often results in QR codes being captured at oblique angles, introducing perspective distortion. The dataset also encompasses QR codes of varying sizes and contrast levels, reflecting real-world usage.

### 4.1.2 METRICS

To evaluate the performance of our QR code reading system and compare it with other available libraries, we employed the success rate metric, which measures the proportion of successfully decoded QR codes out of the total number available codes in the dataset. This metric provides a comprehensive assessment of our system's ability to accurately read QR codes and encompasses both detection and decoding performances, allowing for direct comparison with other methods.

## 4.2 EXPERIMENTAL RESULTS

The success rates of the individual steps and the overall system are presented in Table 1. We found that the majority of decoding failures were due to small and dense codes. Intuitively, it can be observed from Figure 7 that it is relatively easier to decode larger codes than smaller ones as the success rate increases among larger code patches. For example, the decoding success rate among code sizes greater than $100 \times 100$ sq. pixels is over 79%, which should significantly increase the end-to-end success rate.

| Processing Stage | Success Rate (%) |
|------------------|------------------|
| Detection        | 94.40            |
| Decoding         | 70.82            |
| End-to-End       | 66.86            |

Table 1: Success rates at various stages of our processing pipeline. The decoding success rate is reported as a fraction of codes successfully decoded among the detected codes, to isolate the performance of each step. The end-to-end success rate considers the fractions of codes that were successfully detected as well decoded among all the codes in our data set.

Thus, for challenging small-code scenarios, we propose a strategy to fail gracefully while improving user-experience by providing appropriate user-feedback. For example, if the detected code is too small to be decoded successfully, we can send a message indicating that the code cannot be read and suggest retaking the image at a closer distance. If the user has already tried multiple times and continues to fail, we can provide different messaging to indicate that the code is too small to read and suggest alternative methods for scanning the code.

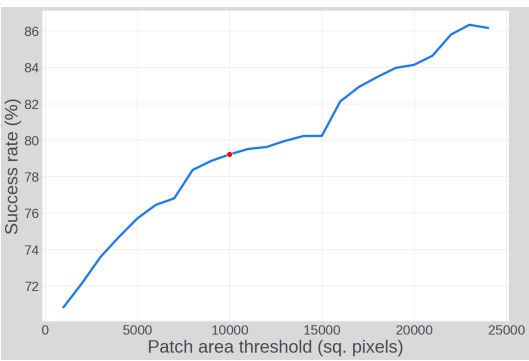

Figure 7: Decoding success rate (%) vs code size (square pixels). Red dot indicates a $100 \times 100$ pixel code patch threshold.

In addition to small and dense codes, other opportunities to improve our approach include partially blocked, peculiarly shaped, and low-quality codes. Examples of these scenarios are presented in Figure 8. Of these, the peculiarly shaped codes are primarily a challenge for the detection model that can be potentially remedied with introducing diversity of code shapes and styles into the training set. Decoding improvements on the rest of the challenging scenarios are an open technical challenge that meanwhile needs to be addressed as a user-experience undertaking.

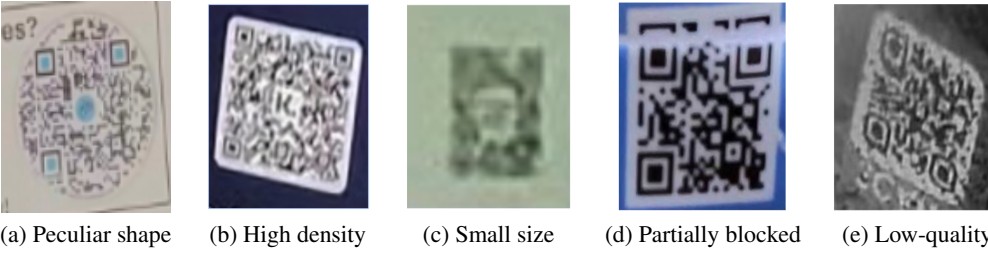

(a) Peculiar shape    (b) High density    (c) Small size    (d) Partially blocked    (e) Low-quality

Figure 8: Examples of challenging codes

### 4.3 COMPARATIVE ANALYSIS

We conduct comprehensive comparative QR code quality analysis on internally collected, ego-centric data for our QR code stack and off-the-shelf QR code pipelines. The internally collected dataset consisted of approximately 528 ego-centric images, with each image having 1 or more QR codes of different size. We define a successful reading as QR code being successfully detected and

Table 2: Comparative analysis table.

| Method | # Successful Readings | Success Rate(%) | Support Multiple QR Code |
|---|---|---|---|
| zxing[zxing] | 118 | 17(%) | N |
| pyzbar[zbar] | 292 | 42(%) | Y |
| qreader[qreader] | 293 | 42(%) | Y |
| WeChat[wechat] | 309 | 44(%) | Y |
| dynamsoft[dynamsoft] | 345 | 50(%) | Y |
| Ours without SR | 444 | 64(%) | Y |
| Ours with SR | 462 | 66 (%) | Y |

decoded. The success rate is defined as number of successful readings divided by number of QR codes. Table 2 shows the results of our runtime v.s. several off-the-shelf QR scan pipeline. By leveraging a light-weight super resolution model, we further improve the capability to scan extremely small QR code and lift the overall absolute scan success rate by 2%. Overall at egocentric setting, our relative scan success rate is at least 34% higher than the best off-the-shelf solution.

## 5  CONCLUSION AND PRACTICAL IMPLICATIONS

In this paper, we presented a novel, lightweight architecture for reading QR codes in egocentric settings, specifically designed to run on wearable devices with minimal battery and latency impacts. Our approach addresses the unique challenges posed by wearable devices, including limited computational resources, single-shot image capture, and the complexities of egocentric imagery such as potential motion blur and wider field of view. Our key contributions include the development of an efficient QR code detection component that can quickly locate potential QR codes within high-resolution egocentric images, and an enhanced decoding component that successfully extracts and interprets encoded information from QR codes captured in challenging egocentric conditions. When evaluated on a comprehensive dataset of egocentric, high-resolution images, our algorithm demonstrates a 34% improvement in QR code reading compared to existing state-of-the-art readers. These advancements pave the way for more seamless integration of QR code functionality in wearable devices, enhancing user experience across various applications, from augmented reality to accessibility assistance.

The rising prominence of multi-modal AI models Bordes et al. (2024) is reshaping our interaction with the world, making it increasingly seamless. In this context, the ability to efficiently read and interpret QR codes becomes important. QR codes serve as a bridge between the physical and digital worlds, providing quick access to additional information that can enrich AI-driven experiences. Consider a scenario where a wearable camera user encounters a restaurant menu presented as a QR code, a trend that has gained popularity, especially in the wake of recent global health concerns. Our enhanced QR code reading capability could allow the AI model to quickly scan and interpret the menu, potentially offering personalized recommendations based on the user's dietary preferences or restrictions, without the need for manual input or smartphone interaction. In scenarios where minimizing physical contact is preferred, such as public transportation or retail environments, hands-free QR code scanning offers a hygienic alternative to touchscreen interactions.

While our work represents a significant step forward in QR code reading for wearable devices, several avenues for future research and development remain. Leveraging multiple frames to improve QR code reading ability could enhance performance, especially in challenging lighting or motion conditions. This approach could involve techniques such as super-resolution or temporal noise reduction. Addressing the challenges posed by small QR codes remains important, with future work focusing on improving both detection and decoding of these codes, particularly when pixel contrast diminishes due to size constraints. Developing better detection models these problematic codes and provide appropriate feedback to the user is essential. Implementing a system that can take multiple pictures with different camera settings when a user attempts to scan a QR code could significantly improve success rates. This might involve rapid adjustment of exposure, focus, or other parameters to optimize image quality for QR code reading. Further research into seamlessly integrating QR

code information with other sensory inputs and AI models could lead to more sophisticated and context-aware applications. Continued optimization of the algorithm to reduce power consumption will be important for long-term adoption in battery-constrained wearable devices.

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
