# OpenReview forum: "EgoQR: Efficient QR Code Reading in Egocentric Settings"
_ICLR.cc/2025/Conference — Submitted to ICLR 2025_

### Official Review · Reviewer_7scQ · 2024-11-02

**Soundness:** 1
**Presentation:** 1
**Contribution:** 1
**Rating:** 3
**Confidence:** 5

**Summary:**

The paper presents a system for QR code detection and recognition in egocentric settings. The authors combine existing tools and techniques, including Faster R-CNN for detection and various image enhancement techniques for decoding, along with a disambiguation method for multiple QR codes. While the paper addresses an interesting practical problem, it has several significant limitations that make it unsuitable for publication at ICLR.

**Strengths:**

The paper addresses a practical and relevant problem, as QR code reading in egocentric/wearable settings is becoming increasingly important with the rise of AR/VR devices.

**Weaknesses:**

**Note that this paper is over-paged (It should be desk rejected?)**

1. Limited Technical Novelty:
- The paper primarily integrates existing methods (Faster R-CNN, traditional image processing techniques, LRSRN for super-resolution) without introducing significant algorithmic innovations
- The image enhancement pipeline is a straightforward combination of standard techniques (color inversion, multi-scale processing, CLAHE, morphological operations)
- The disambiguation approach relies on existing ROI detection from another system (Lumos)

2. Questionable Experimental Setup:
- The dataset collection methodology lacks rigor and proper controls
- No clear description of what constitutes an "egocentric" image in their data collection
- No comparison with actual egocentric devices or properly collected egocentric datasets
- The baseline comparisons (zxing, pyzbar, etc.) are with general-purpose QR code readers rather than systems specifically designed for egocentric settings

3. Insufficient Validation:
- The paper claims to address egocentric-specific challenges but doesn't provide quantitative analysis of how their system handles these specific issues
- No ablation studies to justify the choice and order of enhancement techniques
- Limited analysis of computational efficiency and power consumption, despite claiming suitability for wearable devices
- The claimed 34% improvement lacks context as it's compared against general-purpose QR readers rather than egocentric-specific solutions

4. Methodological Issues:
- The dataset size (528 images) seems inadequate for comprehensive evaluation
- No clear distinction between training and test sets
- Lack of detailed analysis of failure cases and their relationship to egocentric challenges
- No discussion of statistical significance in the reported improvements

5. Incomplete Resource Analysis:
- Despite targeting wearable devices, there's limited discussion of memory usage and power consumption
- No concrete benchmarks on actual wearable hardware

While the problem is interesting and practically relevant, the paper's limitations in technical novelty and experimental validation make it unsuitable for ICLR. The work would be more appropriate for an applied computer vision or system-focused venue after addressing the identified issues.

**Questions:**

- Could you clarify what specific devices were used for data collection? Was this data collected using actual AR/VR headsets or wearable cameras?
- What was your protocol for ensuring the collected data truly represents egocentric viewpoints and challenges?
- Would you share details about the train/test split and validation methodology?

---

### Official Review · Reviewer_mbaf · 2024-11-03

**Soundness:** 2
**Presentation:** 2
**Contribution:** 2
**Rating:** 3
**Confidence:** 3

**Summary:**

The paper proposes EgoQR, a system designed to enhance QR code reading on smart wearable devices from an egocentric perspective. EgoQR addresses challenges like code distortion, varying perspectives, and resource constraints through efficient detection and decoding components tailored for egocentric images. Evaluations on a self-collected dataset show a 34% improvement over existing QR code readers.

**Strengths:**

1. The paper identifies a practical problem: the growing need for efficient and frictionless QR code reading capabilities from egocentric viewpoints, particularly with the rising adoption of smart wearable devices with limited computational resources.

2. The paper introduces a solution to tackle issues like code distortion and resource constraints. Experimental results on a self-collected dataset demonstrate its superior performance compared to other commercial applications.

**Weaknesses:**

1. The novelty of the paper is limited, resembling more of a technical report addressing an engineering problem than a formal academic paper. The core modules of the proposed pipeline are nearly all previous methods.

2. The experimental section is insufficient, as it only reports the accuracy of different methods without providing in-depth analysis or conducting an ablation study on various components.

3. The paper lacks a comparison of the public dataset.

**Questions:**

No.

---

### Official Review · Reviewer_yZcX · 2024-11-04

**Soundness:** 3
**Presentation:** 3
**Contribution:** 2
**Rating:** 3
**Confidence:** 3

**Summary:**

This work introduces a system for reading QR codes from first-person perspectives on wearable devices. It tackles challenges like wide field-of-view and code distortion using a detection and decoding approach that includes a tailored Faster R-CNN model and image enhancement techniques. The system has a 34% improvement in QR code reading over existing methods on a novel egocentric dataset. Future work will focus on improving the detection and decoding of small QR codes and integrating the system with other AI models for more context-aware applications.

**Strengths:**

-  The proposed EgoQR system demonstrated a significant 34% improvement in QR code reading success over existing state-of-the-art readers.
- By incorporating image enhancement techniques like super-resolution and adaptive histogram equalization, the system improves decoding success rates even in challenging conditions such as motion blur and varying lighting.

**Weaknesses:**

- Lack of novelty: The paper builds upon well-established techniques such as Faster R-CNN for detection and common image enhancement methods for decoding.
- Concern about efficiency: The proposed system employs high-resolution image processing and machine learning models which could be computationally intensive. It would be beneficial to see a more detailed analysis of the trade-offs between performance and resource consumption.

**Questions:**

It would be helpful if the authors provide explanations on the novelty of this system. Also, I hope the authors provide experiments to compare the efficiency of the proposed system against existing approaches.

---

### Official Review · Reviewer_ePrs · 2024-11-04

**Soundness:** 2
**Presentation:** 1
**Contribution:** 1
**Rating:** 1
**Confidence:** 4

**Summary:**

This paper proposes a QR code processing system that consists of QR code detection, an image enhancement module, and QR code decoding. The paper claims two challenges in QR code reading: 1) Stylistic QR code and 2) Small QR code. The proposed method resolves the issue by utilizing image enhancement modules, such as color inversion, multi-scale processing, contrast enhancement, morphological operation, and super-resolution. As a result, the method achieved a higher success rate compared to previous off-the-shelf QR scan methods. However, the main weakness of the paper is novelty. The paper is a naive systematic paper that combines existing methods to resolve the QR code problem. It isn't sufficient enough for ICLR's novelty standard.

**Strengths:**

S1. The proposed method resolves stylistic QR code and small QR code problems by utilizing image enhancement modules, such as color inversion, multi-scale processing, contrast enhancement morphological operation, and super-resolution.

**Weaknesses:**

W1. Novelty.
* The paper proposes a simple pipeline that combines a QR code detection module, an image enhancement module, and a QR code decoding module. It is difficult to find any new idea, novelty, or new perspective for each component. ICLR requires new and brilliant ideas, perspectives, and contributions to various research fields. However, the paper's contribution is not enough to meet ICLR standards.

W2. Presentation
* The paper presentation needs to be clarified. For instance, Table 2 doesn't properly reference each method.
Also, Fig 8 is mentioned earlier than Fig 1. The figure and table should be mentioned in their order.

W3. Experiments
* The effect of each image enhancement method is unclear. For instance, the stylistic QR code problem is resolved by introducing a number of enhancement tools, such as color inversion, multi-scale processing, contrast enhancement, and morphological operation. Are they all necessary? How do they affect the performance of stylistic QR code reading?

**Questions:**

Please see weakness part.

**Details Of Ethics Concerns:**

None.

---

### Official Review · Reviewer_gQQi · 2024-11-06

**Soundness:** 3
**Presentation:** 3
**Contribution:** 2
**Rating:** 1
**Confidence:** 5

**Summary:**

This paper proposes a QR code reader for wearable devices. The Implementation of this system incorporates existing modules such as Faster-RCNN and ZXing. In addition, a series of image processing algorithms are used for enhancing decoding rate. Overall, the practical significance of this paper is greater than its algorithmic innovation, and it is not suitable for submission to conferences such as ICLR, which emphasize theoretical innovation or exploration. We recommend that the authors submit this paper to a conference in the field of embodied intelligence.

**Strengths:**

1.complex system implementation
2.great practical significance
3.experimental results in real-world environments

**Weaknesses:**

1.The method lacks novelty
2.The method is a simple combination of existing modules

**Questions:**

1.	The core modules of existing QR reader are locating and alignment, how to solve these two problems in this paper? The authors use faster-rcnn to detect QR codes, however faster-rcnn can only predict horizontal bounding box, thus the output roi suffer perspective deformation.
2.	Faster-rcnn is a two-stage model, which may be time-consuming in edge-ai device, why not use yolo? The newest version yolo has support well for small object detection.
3.	For question#1, why not use keypoint detection to replace the object detection? If you can determine the vertices of qr code, the perspective deformation can be removed by compute homograph matrix, which can play a role of alignment.
4.	Please explain the implementation environments.
5.	Table 2 doesn’t show running time, which is important for edge-ai.

---

### Meta-Review · Area_Chair_yUgx · 2024-12-23

**Metareview:**

The paper introduces EgoQR, a system for efficient QR code reading on smart wearable devices in egocentric settings. It addresses challenges like code distortion, wide field-of-view, and resource constraints with a detection component using Faster R-CNN and decoding enhanced by image processing techniques (e.g., super-resolution, contrast adjustment). Evaluated on a self-collected egocentric dataset, EgoQR achieves a 34% improvement in QR code reading success over existing methods. While the approach is practical and effective, its novelty lies in the system integration rather than algorithmic innovation.

Strengths:
Practical Relevance: Addresses QR code reading in egocentric settings, critical for wearable and AR/VR devices.
Performance Boost: Achieves 34% better QR code reading success than state-of-the-art methods.
Robustness: Handles challenges like small, distorted codes with advanced image enhancements.
Efficient System: Proposes a resource-efficient solution validated in real-world scenarios.

Weaknesses:
Limited Novelty: Combines existing methods (e.g., Faster R-CNN, standard image processing) without significant innovation.
Weak Experiments: Lacks ablation studies, detailed analysis, and comparisons with egocentric-specific solutions.
Methodological Gaps: Small dataset, unclear train-test split, and insufficient failure case or statistical analysis.

We reject the paper due to limited novelty on the method.

**Additional Comments On Reviewer Discussion:**

No rebuttal

---

### Decision · Program_Chairs · 2025-01-22

Reject